

# Bcl-2 homologue Debcl enhances α-synuclein-induced phenotypes in Drosophila

P. Githure M'Angale and Brian E. Staveley

Department of Biology, Memorial University of Newfoundland, St. John's, Newfoundland and Labrador, Canada

## ABSTRACT

**Background:** Parkinson disease (PD) is a debilitating movement disorder that afflicts 1–2% of the population over 50 years of age. The common hallmark for both sporadic and familial forms of PD is mitochondrial dysfunction. Mammals have at least twenty proapoptotic and antiapoptotic *Bcl-2* family members, in contrast, only two *Bcl-2* family genes have been identified in *Drosophila melanogaster*, the proapoptotic mitochondrial localized *Debcl* and the antiapoptotic *Buffy*. The expression of the human transgene α-*synuclein*, a gene that is strongly associated with inherited forms of PD, in dopaminergic neurons (DA) of Drosophila, results in loss of neurons and locomotor dysfunction to model PD in flies. The altered expression of *Debcl* in the DA neurons and neuron-rich eye and along with the expression of α-*synuclein* offers an opportunity to highlight the role of *Debcl* in mitochondrial-dependent neuronal degeneration and death.

**Results:** The directed overexpression of *Debcl* using the *Ddc-Gal4* transgene in the DA of Drosophila resulted in flies with severely decreased survival and a premature age-dependent loss in climbing ability. The inhibition of *Debcl* resulted in enhanced survival and improved climbing ability whereas the overexpression of *Debcl* in the α-*synuclein*-induced Drosophila model of PD resulted in more severe phenotypes. In addition, the co-expression of *Debcl* along with *Buffy* partially counteracts the *Debcl*-induced phenotypes, to improve the lifespan and the associated loss of locomotor ability observed. In complementary experiments, the overexpression of *Debcl* along with the expression of α-*synuclein* in the eye, enhanced the eye ablation that results from the overexpression of *Debcl*. The co-expression of *Buffy* along with *Debcl* overexpression results in the rescue of the moderate developmental eye defects. The co-expression of *Buffy* along with inhibition of *Debcl* partially restores the eye to a roughened eye phenotype.

**Discussion:** The overexpression of *Debcl* in DA neurons produces flies with shortened lifespan and impaired locomotor ability, phenotypes that are strongly associated with models of PD in Drosophila. The co-expression of *Debcl* along with α-*synuclein* enhanced the PD-like phenotypes. The co-expression of *Debcl* along with *Buffy* suppresses these phenotypes. Complementary experiments in the Drosophila eye show similar trends during development. Taken all together these results suggest a role for *Debcl* in neurodegenerative disorders.

Corresponding author
Brian E. Staveley, bestave@mun.ca

## INTRODUCTION

Parkinson disease (PD) is a human movement disorder that is strongly associated with the selective and profound degeneration and loss of dopaminergic (DA) neurons to result in a set of marked clinical features (*Forno, 1996*). The neuropathological hallmarks exhibited by PD patients include the presence of Lewy Bodies (LB) which are intracytoplasmic proteinaceous inclusions composed of α-*synuclein* and ubiquitin among other proteins (*Forno, 1996*; *Leroy et al., 1998*; *Polymeropoulos et al., 1997*). This atypical protein aggregation and accumulation is believed to lead to cellular toxicity and contribute to the pathogenesis of PD. Additional pathological mechanisms that are associated with PD include aberrant protein aggregation and mitochondrial damage (*Gupta, Dawson & Dawson, 2008*; *Schulz, 2007*; *Whitworth, 2011*). Familial forms of PD have highlighted the genetic basis of PD and the study of the associated gene loci in model organisms offers great understanding of the disease aetiology and pathology (*Ambegaokar, Roy & Jackson, 2010*; *Gasser, 2009*; *Guo, 2012*). The gene encoding α-*synuclein*, a small soluble protein of largely unknown function predominantly found in neural tissues, was first to be identified as responsible for inherited PD (*Polymeropoulos et al., 1997*). Mitochondrial dysfunction due to the accumulation of α-*synuclein* has been implicated as one of the mechanisms leading to PD (*Chinta et al., 2010*; *Choubey et al., 2011*; *Esteves et al., 2011*; *Zhu et al., 2011*). The association of α-*synuclein* with components of the mitochondria is thought to lead to oxidative stress, apoptosis, autophagy and eventually, neurodegeneration. The first Drosophila model of PD utilized a human α-*synuclein* transgene to induce the PD-like symptoms (*Feany & Bender, 2000*). This model system is very successful and widely applied, as it displays the age-dependent loss of locomotor function, the degeneration of DA neurons and LB-like inclusions, features that are present in human PD (*Auluck et al., 2002*; *Botella et al., 2009*; *Büttner et al., 2014*; *Feany & Bender, 2000*; *Kong et al., 2015*; *Staveley, 2014*; *Webb et al., 2003*; *Zhu et al., 2016*). Drosophila has available tissue specific gene enhancers such as *TH-Gal4*, *elav-Gal4* and *Ddc-Gal4*, which are used to model PD in flies in combination with the powerful bipartite UAS/Gal4 (*Brand & Perrimon, 1993*) system. Of importance is the correlation between DA neuron loss and the age-dependent loss of locomotor function (*Park, Schulz & Lee, 2007*; *Staveley, 2014*) which validates the implication that age-dependent loss of locomotor function is as a result of DA neuron degeneration.

The *Bcl-2* family of genes are crucial controllers of apoptosis in animals and are functionally composed of proapoptotic and antiapoptotic members (*Adams & Cory, 1998*; *Cory & Adams, 2002*; *Fu & Fan, 2002*; *Siddiqui, Ahad & Ahsan, 2015*). In mammals, this multigene family has about 20 members, the antiapoptotic proteins protect the mitochondria from disruption by the proapoptotic proteins (*Colin et al., 2009*; *Cory & Adams, 2002*; *Martinou & Youle, 2011*; *Suen, Norris & Youle, 2008*; *Tsujimoto, 2002*). The antiapoptotic members possess four *Bcl-2* homology (BH) domains while the proapoptotic members have three to four BH domains. The proapoptotic proteins initiate apoptosis by

the permeabilization of the outer mitochondrial membrane which results in the release of apoptogenic factors into the cytosol (*Delbridge & Strasser, 2015*; *Doerflinger, Glab & Puthalakath, 2015*; *Li & Dewson, 2015*; *Lopez & Tait, 2015*). The antiapoptotic members protect the mitochondria from permeabilization by the proapoptotic members and block the release of apoptogenic factors such as cytochrome c, apoptosis inducing factor (AIF) among others from being released from the inner mitochondrial membrane into the cytosol.

*Drosophila melanogaster* possesses many of the apoptotic pathway proteins that participate in the intrinsic and extrinsic cell death pathways (*Kornbluth & White, 2005*; *Richardson & Kumar, 2002*). The *Bcl-2* family member homologues in Drosophila are limited to the single antiapoptotic *Buffy* (*Quinn et al., 2003*), and the sole proapoptotic *death executioner Bcl-2* homologue, *Debcl* (*Brachmann et al., 2000*; *Colussi et al., 2000*; *Igaki et al., 2000*; *Quinn et al., 2003*; *Zhang et al., 2000*). *Debcl* has a strong similarity with the mammalian mitochondria outer membrane permeabilization protein Bok/Mtd.

The promoter region of *Debcl* contains four dNF-Y-binding consensus sequences which play positive roles in promoter activity and indicate that dNF-Y regulates *Debcl* gene expression (*Ly et al., 2013*). The tumour suppressor gene *Retinoblastoma* (*Rbf1* in Drosophila) induces a *Debcl-*and *Drp1*-dependent mitochondrial cell death (*Clavier et al., 2015*). *Rbf1* induces cell death by reducing the expression of the sole *Debcl* antagonist *Buffy* (*Clavier et al., 2014*). The *Rbf1*-induced apoptosis is dependent on *Debcl*-dependent mitochondrial ROS production and essentially *Debcl* is required downstream of *Buffy* for apoptosis to occur. The *Debcl*-induced ROS production appears to be through Glycerophosphate oxidase 1 participation to increase mitochondria ROS accumulation (*Colin et al., 2015*). The organic solute carrier partner 1/oxidored nitrodomain-containing protein 1 (OSCP1/NOR1), a known tumour suppressor induces apoptosis by the down-regulation of the *Buffy* gene and the up-regulation of the *Debcl* gene (*Huu, Yoshida & Yamaguchi, 2015*). *Debcl* is not required for most developmental cell death, but has been shown to play a role in embryonic cell death (*Galindo et al., 2009*) and stress-induced apoptosis (*Sevrioukov et al., 2007*). Antiapoptotic *Buffy* antagonizes *Debcl*-induced apoptosis by physical interaction (*Quinn et al., 2003*), probably at the mitochondria where *Debcl* localizes (*Doumanis, Dorstyn & Kumar, 2007*). The presence of a mitochondrial outer membrane (MOM)-targeting motif in *Debcl* indicates it possibly has a role in mitochondrial cell death pathway.

The role of the mitochondria in PD pathogenesis makes the α-*synuclein*-induced model of PD (*Feany & Bender, 2000*) a very attractive model for the investigation of the role of *Bcl-2* proteins. Here, we investigate the potential enhancement or suppression of the α-*synuclein*-induced PD phenotypes by the inhibition and overexpression of the pro-apoptotic *Bcl-2* homologue *Debcl*.

# MATERIALS AND METHODS

## Drosophila media and culture

Stocks and crosses were maintained on standard cornmeal/molasses/yeast/agar media treated with propionic acid and methylparaben. Stocks were sustained on solid media for two to three weeks before being transferred onto new media to re-culture. Stocks were

kept at room temperature (22 ± 2 °C) while crosses and experiments were carried out at 25 and 29 °C.

## Drosophila stocks and derivative lines

*UAS-debcl, UAS-Buffy* (*Quinn et al., 2003*) were a gift from Dr. Leonie Quinn of University of Melbourne, *UAS-α-synuclein* (*Feany & Bender, 2000*) by Dr. M. Feany of Harvard Medical School and *Ddc-Gal4* (*Li et al., 2000*) by Dr. J. Hirsch of University of Virginia. *y¹ v¹; P(y(+t7.7) v(+t1.8) = TRiP.JF02429)attP2* was developed at Harvard Medical School (http://fgr.hms.harvard.edu/trip-vivo-rnai-approach) (*Perkins et al., 2015*) hereby referred to as *UAS-Debcl-RNAi*, *GMR-Gal4* (*Freeman, 1996*) and *UAS-lacZ* were sourced from the Bloomington Drosophila Stock Center at Indiana University. The *UAS-α-synuclein Ddc-Gal4/CyO; UAS-α-synuclein GMR-Gal4/CyO; UAS-Buffy Ddc-Gal4//CyO* and *UAS-Buffy GMR-Gal4/CyO* derivative lines were generated using standard homologous recombination methods that have previously been described (*Rong & Golic, 2000*). Briefly, virgin females of *UAS-α-synuclein* or *UAS-Buffy* were crossed to males bearing the *Gal4* transgene of interest (*Ddc-Gal4* or *GMR-Gal4*). In the next generation, virgin female progeny were collected and mated to a balancer line (*L/CyO*), then individual male progeny are collected and mated to females of the balancer line. The resulting progeny was then mated and the homozygotes arising from the next generation was selected and tested by PCR. The derivative lines were used for the overexpression of either *α-synuclein* or *Buffy* along with selected transgenes 1) in *Ddc-Gal4*-expressing neurons that include the DA neurons or 2) in the developing eye behind the morphogenetic furrow directed by the *GMR-Gal4* transgene. PCR reactions and gel electrophoresis were used for analysis of recombination events. PCR reaction was used to determine the amplification of DNA products from primers designed from the *Homo sapiens* synuclein, alpha (non A4 component of amyloid precursor) (SNCA), transcript variant 1 mRNA, NCBI reference sequence: NM_000345.3 using the NCBI primer design tool. The 5′ to 3′ sequence of the forward primer was GTGCCCAGTCATGACATTT, while that of the reverse primer was CCACAAAATCCACAGCACAC and were ordered from Invitrogen. The *Drosophila melanogaster Buffy* mRNA, NCBI reference sequence: NM_078978.2, was used to design a set of *Buffy* primers that would target both the endogenous and the overexpression transcripts. The 5′ to 3′ sequence of the forward primers were CACAGCGTTTATCCTGCTGA and CGGGTGGTGAGTTCCATACT, while that of the reverse primers were TCGCAGTGTGAAGATTCAGG and TTAATCCACGGAACCAGCTC, and were ordered from Eurofins MWG Operon. Gel electrophoresis was used for confirmation of recombination events via presence of the PCR product.

## Ageing assay

Several single vial matings were made and a cohort of critical class male flies was collected upon eclosion. At least 200 flies were aged per genotype at a density of 20 or fewer flies per vial to avoid crowding on fresh media which was replenished every other day.

Flies were observed and scored every two days for the presence of deceased adults. Flies were considered dead when they did not display movement upon agitation (*Staveley, Phillips & Hilliker, 1990*). Longevity data was analysed using the GraphPad Prism version 5.04 and survival curves were compared using the log-rank (Mantel-Cox) test. Significance was determined at 95%, at a P-value less than or equal to 0.05 with Bonferroni correction.

## Climbing assay

A batch of male flies was collected upon eclosion and scored for their ability to climb (*Todd & Staveley, 2004*). Every seven days, 50 males from every genotype were assayed for their ability to climb 10 cm in 10 s in a clean climbing apparatus in 10 repetitions. Analysis was performed using GraphPad Prism version 5.04 and climbing curves were fitted using non-linear regression and compared using 95% confidence interval with a 0.05 P-value.

## Scanning electron microscopy of the Drosophila eye

Several single vial crosses were made at 29 °C and adult male flies collected upon eclosion and aged for three days before being frozen at −80 °C. Whole flies were mounted on scanning electron microscope stubs, desiccated overnight and photographed with a FEI Mineral Liberation Analyzer 650F scanning electron microscope. For each cross at least 10 eye images were analysed using the National Institutes of Health (NIH) ImageJ software (*Schneider, Rasband & Eliceiri, 2012*) and biometric analysis performed using GraphPad Prism version 5.04. The percent area of eye disruption was calculated as previously described (*M'Angale & Staveley, 2012*).

## RESULTS

### *Debcl* is similar to the human proapoptotic *Bcl-2 ovarian killer (Bok)*

Bioinformatic analysis of the protein sequences encoded by the *Debcl* and *Bok* genes reveal 37% identity and 55% similarity. The *Debcl* protein consists of 300 amino acids and indicates the existence of the BH1, BH2, BH3, BH4 and TM domains, similar to the 212 amino acids human Bok (Fig. 1). An ELM resource search for functional sites (*Dinkel et al., 2016*) indicates the presence of a transmembrane domain (membrane anchor region), an inhibitor of apoptosis binding motif (IBM) at amino acids 1–5, a PDZ domain at amino acids 295–300, an ER retention motif at amino acids 109–115 and between amino acids 258–262, an Atg8 binding motif at amino acids 36–42, a nuclear receptor box motif at amino acids 295–300, and a ubiquitination motif of the SPOP-binding consensus at amino acids 2–6 and another one at position 74–79. There is a number of BH3-homology region binding sites in the central region of the protein as determined by an NCBI conserved domain search (*Marchler-Bauer et al., 2015*). Although the two proteins Bok and *Debcl* have been determined to be antiapoptotic, both show the presence of a BH4 domain, the homology domain that is most often associated with pro-survival proteins.

**Figure 1 Debcl is related to human Bcl-2 ovarian killer (Bok).** When *debcl* protein is aligned with human Bok the *Bcl-2* homology (BH) domains show strong conservation. Clustal Omega multiple sequence alignment (*Goujon et al., 2010*; *Sievers et al., 2011*) of *Drosophila melanogaster debcl* protein (Dmel is *Drosophila melanogaster* NP_788278.1) with the human Bok (Hsap is *Homo sapiens* NP_115904.1), mouse Bok (Mmus is *Mus musculus* NP_058058.1) and mosquito Bok (Agam is *Anopheles gambiae* XP_309956.4) showing the highlighted conserved BH domains and the TM helices. The domains were identified using NCBI Conserved Domain Database Search (CDD) (*Marchler-Bauer et al., 2015*) and ELM resource search for functional sites (*Dinkel et al., 2016*). "*" indicate the residues that are identical, ":" indicate the conserved substitutions, "." indicate the semi-conserved substitutions. Colours show the chemical nature of amino acids. Red is small hydrophobic (including aromatic), Blue is acidic, Magenta is basic, and Green is basic with hydroxyl or amine groups.

## Directed misexpression of *Debcl* in DA neurons alters lifespan and locomotor ability

The inhibition of *Debcl* in the DA neurons by RNA interference results in a lifespan with a median survival of 64 days that is similar to 62 days for the controls expressing the benign *lacZ* transgene as determined by a Log-rank (Mantel-Cox) test (Fig. 2A). The locomotor ability showed a slight improvement when nonlinear fitting of the climbing curves was performed, with significant differences at 95% confidence intervals (Fig. 2B).

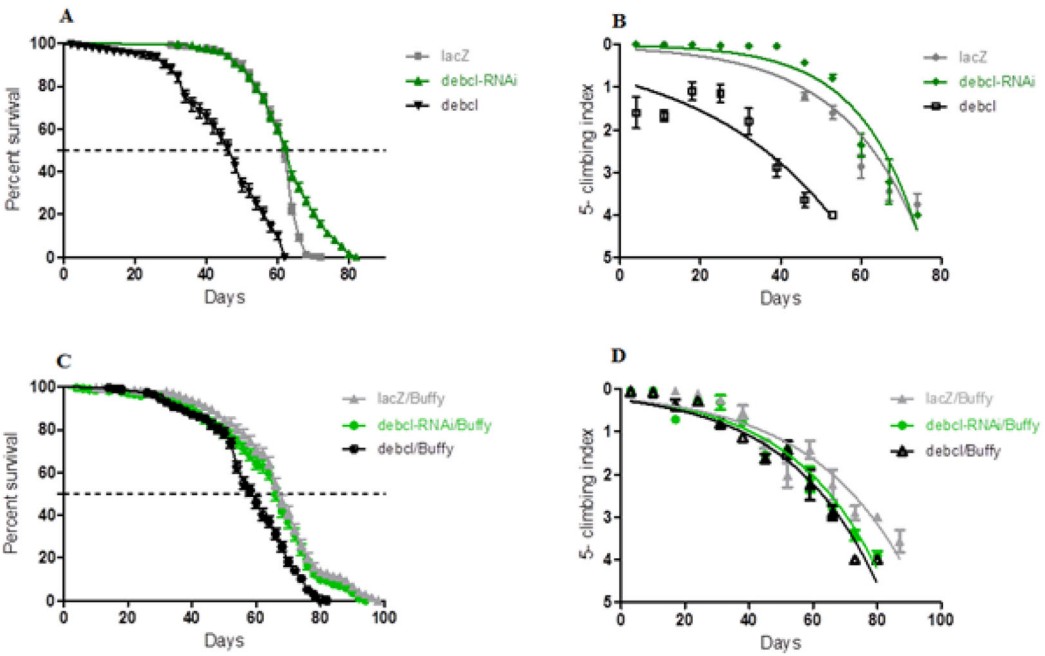

**Figure 2 *Debcl*-induced phenotypes are rescued by the pro-survival *Buffy*.** (A) The directed inhibition of *debcl* in the DA neurons driven by *Ddc-Gal4* results in a slightly increased median survival compared to the control flies overexpressing *UAS-lacZ*, while the overexpression of *debcl* results in severely reduced survival. The genotypes are *UAS-lacZ/Ddc-Gal4; UAS-debcl-RNAi/Ddc-Gal4* and *UAS-debcl/Ddc-Gal4*. Longevity is shown as percent survival (P < 0.01, determined by log-rank and n ≥ 200). (B) The inhibition of *debcl* results in improved climbing ability whereas the overexpression of *debcl* results in a highly compromised climbing ability as determined by non-linear fitting of the climbing curves and comparing at 95% confidence intervals. The genotypes are *UAS-lacZ/Ddc-Gal4; UAS-debcl-RNAi/Ddc-Gal4* and *UAS-debcl/Ddc-Gal4*. Error bars indicate the standard error of the mean (SEM) and n = 50. (C) The overexpression of *Buffy* along with the overexpression of *debcl* or *debcl-RNAi* restores lifespan and (D) significantly improves the climbing ability of these flies. The genotypes are *UAS-Buffy; Ddc-Gal4/UAS-lacZ, UAS-Buffy; Ddc-Gal4/UAS-debcl-RNAi* and *UAS-Buffy; Ddc-Gal4/UAS-debcl*. Longevity was determined by log-rank (Mantel-Cox) test and n ≥ 200 while climbing ability curves were fitted non-linearly and compared with 95% CI.

This suggests that the inhibition of the proapoptotic *Debcl* confers a small advantage for the normal functioning of DA neurons.

When *Debcl* is overexpressed in DA neurons, the survival criteria of these flies differ greatly (Fig. 2A), with *Debcl*-overexpressing flies having a median lifespan of 48 days compared to 62 days for the controls expressing the benign *lacZ* transgene as indicated by a Log-rank (Mantel-Cox) test. The overexpression of *Debcl* in DA neurons severely impairs climbing ability as determined by the nonlinear fitting of the curve with 95% CI (Fig. 2B). This suggests that the overexpression of *Debcl* in DA neurons interferes with the normal functioning of these flies and results in compromised "healthspan."

## The overexpression of the pro-survival *Buffy* rescues the *Debcl*-induced phenotypes

The overexpression of *Buffy* and *Debcl* in DA neurons results in a longer lifespan and improved locomotor ability (Fig. 2). The median lifespan of these flies was 62 days when

compared to *Buffy* and *lacZ* overexpressing controls at 68 days. The median survival of *Debcl-RNAi* flies was 68 days as determined by a Log-rank (Mantel-Cox) test (Fig. 2C). The climbing ability of these flies was also much improved as determined by comparing the climbing indices at 95% CI (Fig. 2D). Taken together these results suggest that *Buffy* antagonizes the *Debcl*-induced phenotypes of shortened lifespan and poor climbing ability to markedly improve "healthspan."

### Altered expression of *Debcl* influences the $\alpha$-*synuclein*-induced phenotypes

The inhibition of *Debcl* by RNAi along with the expression of $\alpha$-*synuclein* under the direction of the *Ddc-Gal4* transgene results in increased lifespan and healthier climbing ability compared to the control (Fig. 3). The *Debcl-RNAi* along with $\alpha$-*synuclein*-expressing flies had a median lifespan of 67 days, while that of $\alpha$-*synuclein*-expressing controls was 60 days as determined by a Log-rank (Mantel-Cox) test (Fig. 3A). The climbing ability of these flies was slightly improved than of the $\alpha$-*synuclein*-expressing controls as indicated by the nonlinear fitting of the climbing curves and compared the 95% CI (Fig. 3B). These results show that the inhibition of the proapoptotic *Debcl* confers a significant advantage to flies under the influence of the neurotoxic effects of the human transgene $\alpha$-*synuclein*.

The overexpression of *Debcl* along with $\alpha$-*synuclein* in DA neurons results in decreased median lifespan of 44 days, compared to 60 days for the control flies as determined by a Log-rank (Mantel-Cox) test (Fig. 3A). The climbing curves indicate that there was a significant reduction in the climbing ability of the flies with overexpression of *Debcl* (Fig. 3B) and thus, enhancing the phenotypes observed when $\alpha$-*synuclein* is expressed in DA neurons. This suggests that the overexpression of *Debcl* further increases the toxic effects of the expression of $\alpha$-*synuclein*.

### Overexpression of *Debcl* enhances the $\alpha$-*synuclein*-induced developmental eye defects

The overexpression of *Debcl* in the Drosophila eye results in severe ablation of the eye due to apoptosis (*Colussi et al., 2000*; *Igaki et al., 2000*) while expression of $\alpha$-*synuclein* in the eye results in developmental defects (Fig. 4D). When *Debcl* is overexpressed in the eye, developmental defects resulting from *Gal4* (*Kramer & Staveley, 2003*) (Figs. 4A and 4J), inhibition of *Debcl* (Figs. 4B and 4J), and overexpression of *Debcl* (Figs. 4C and 4J) are enhanced. Biometric analysis of the ommatidia number and the percentage of eye disruption showed significant differences in the compared genotypes to the control that express the benign *lacZ* transgene (Fig. 4J). The inhibition of *Debcl* along with $\alpha$-*synuclein* expression (Figs. 4E and 4K) and the co-expression of *Debcl* and $\alpha$-*synuclein* (Figs. 4F and 4K) result in enhanced phenotypes. The disruption of the ommatidial array due to fusion of the ommatidia and smaller eye is severely enhanced by the overexpression of *Debcl* together with $\alpha$-*synuclein* (Figs. 4F and 4K). The analysis of the ommatidia number and disruption of the eye reveals significant differences, the inhibition of *Debcl* yields "healthier" eyes and its overexpression results in worsened

A.

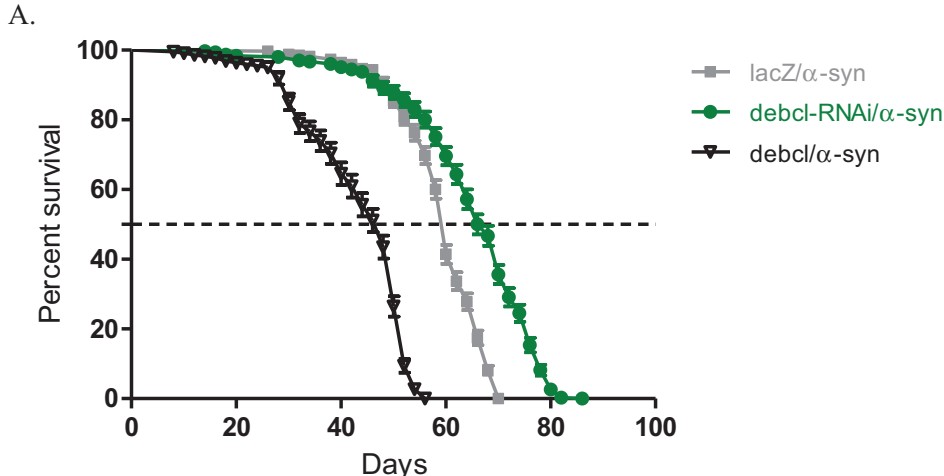

B.

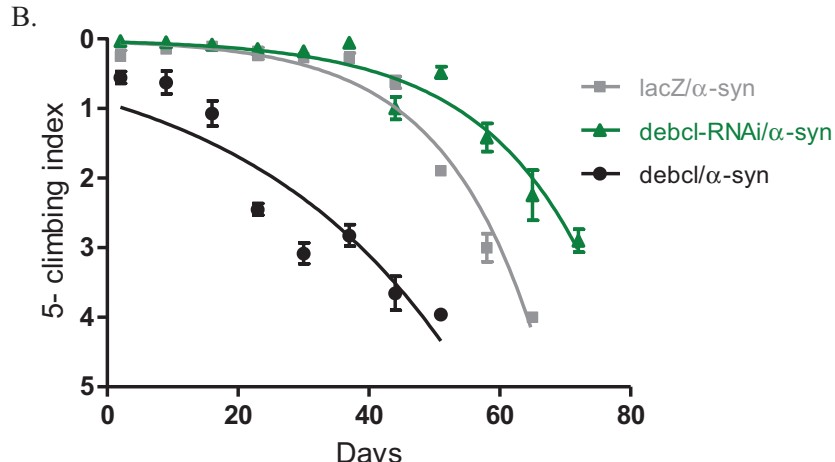

**Figure 3 Overexpression of *debcl* enhances the *α-synuclein*-induced phenotypes.** (A) Directed overexpression of *debcl* in the DA neurons severely decreases longevity whereas its inhibition shows an improvement in lifespan. Genotypes are *UAS-α-synuclein Ddc-Gal4/UAS-lacZ; UAS-α-synuclein Ddc-Gal4/UAS-Debcl-RNAi;* and *UAS-α-synuclein Ddc-Gal4/UAS-Debcl*. Longevity is shown as percent survival (P < 0.01, determined by log-rank and $n \geq 200$). (B) The co-expression of *debcl* in the *α-synuclein* model of PD enhanced the age-dependent loss in climbing ability. The directed inhibition of *debcl* in the DA neurons improved the climbing ability over time compared to the control. The genotypes are *UAS-α-synuclein; Ddc-Gal4/UAS-lacZ, UAS-α-synuclein; Ddc-Gal4/UAS-debcl-RNAi,* and *UAS-α-synuclein; Ddc-Gal4/UAS-debcl*. Analysis of the climbing curves and significance was determined by comparing the 95% confidence intervals. Error bars indicate the SEM and $n = 50$.

phenotypes (Fig. 4K). The ommatidial disarray that results from inhibition of *Debcl* are completely rescued by overexpression of the pro-survival *Buffy* (Figs. 4H and 4L), while the ablated eye that result from *Debcl* overexpression is partially rescued upon *Buffy* overexpression, this restores the eye ablation to a mildly severe rough eye phenotype (Figs. 4I and 4L). Biometric analysis showed recouped ommatidia number and a lessened disruption of the eye, though they were still significantly different from the control (Fig. 4L). These results suggest that overexpression of *Debcl* along with expression

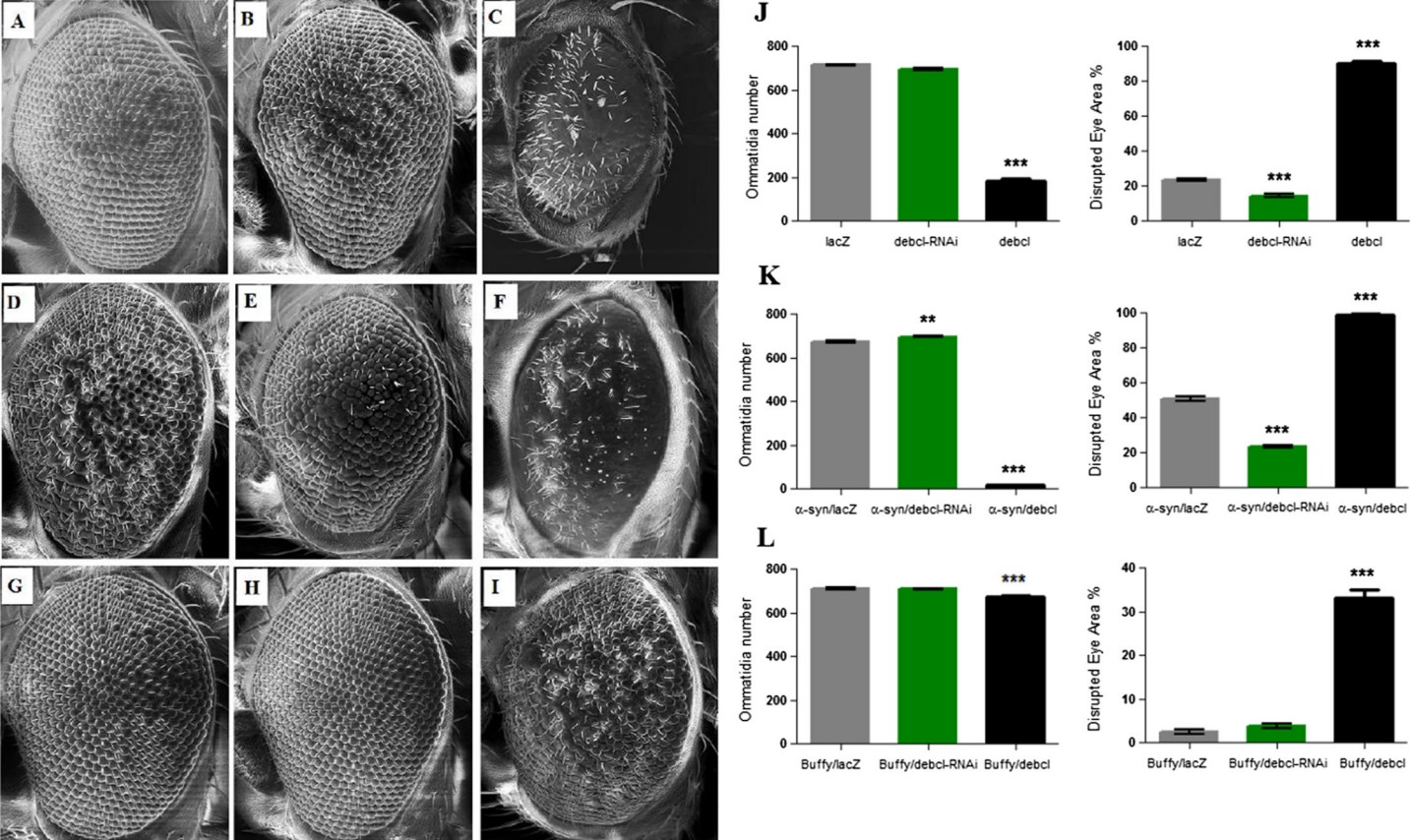

**Figure 4 *Buffy* partially rescues the *Debcl*-induced developmental eye defects.** Scanning electron micrographs when *Debcl* is overexpressed or inhibited in the eye with the eye-specific *GMR-Gal4* transgene; (A) *GMR-Gal4/UAS-lacZ*; (B) *GMR-Gal4/UAS-Debcl-RNAi*; (C) *GMR-Gal4/UAS-Debcl*; when co-expressed with α-synuclein; (D) *UAS-α-synuclein; GMR-Gal4/UAS-lacZ*; (E) *UAS-α-synuclein; GMR-Gal4/UAS-Debcl-RNAi* (F) *UAS-α-synuclein; GMR-Gal4/UAS-Debcl*; and when co-expressed with *Buffy*; (G) *UAS-Buffy; GMR-Gal4/UAS-lacZ* (H) *UAS-Buffy; GMR-Gal4/UAS-Debcl-RNAi* and (I) *UAS-Buffy; GMR-Gal4/UAS-Debcl*. (J) Biometric analysis showed a significant difference in the disrupted area of the eye when *Debcl* is inhibited in the developing eye, and a decreased number of ommatidia and high levels of disruption when *Debcl* is overexpressed. (K) Biometric analysis indicates a marked difference when *Debcl* is inhibited along with the expression of α-synuclein, with increased ommatidia number and a less disrupted ommatidial array, whereas the overexpression of *Debcl* along with the expression of α-synuclein results in a dramatic decrease in ommatidia number coupled with severe ommatidial disarray. (L) The biometric analysis reveals the restoration of *Debcl*-induced phenotypes by overexpression of *Buffy*. The inhibition and overexpression of *Debcl* along with overexpression of *Buffy*, results in increased ommatidia number and improved disruption of the ommatidial array, to produce "healthier" eyes as determined by a one-way ANOVA and Dunnett's multiple comparison test (P < 0.05 and 95% CI), error bars indicate the SEM, asterisks (*) represents statistically significant result and *n* = 10.

of α-synuclein enhances the *Debcl*-induced eye ablation, while the overexpression of *Debcl* together with *Buffy* partially rescues the eye phenotype.

## DISCUSSION

Since mitochondrial dysfunction is central to the pathology of both sporadic and familial forms of PD (*Subramaniam & Chesselet, 2013*), it was important to highlight the role and consequences of the altered expression of the proapoptotic mitochondrial gene *Debcl* in this process. The overexpression of *Debcl* in Drosophila and other systems, including mammalian, has been demonstrated to lead to apoptosis (*Brachmann et al., 2000*; *Colussi et al., 2000*; *Galindo et al., 2009*; *Igaki et al., 2000*; *Senoo-Matsuda, Igaki & Miura, 2005*;

*Sevrioukov et al., 2007*; *Zhang et al., 2000*). The recapitulation of PD-like symptoms in *Drosophila melanogaster*, especially the age-dependent loss of climbing ability, has led to investigation of genes that could suppress these phenotypes (*Auluck et al., 2002*; *Feany & Bender, 2000*; *Haywood & Staveley, 2004*). Our results show that the overexpression of *Debcl* results in a severely shortened lifespan followed by premature loss in climbing ability; phenotypes that are reminiscent of PD-like symptoms in model organisms. Thus our work shows the intricate balance between life and death decisions in the sensitive dopamine producing neurons. It seems that excess amounts of *Debcl* protein are sufficient to upset the survival mechanisms and lead to degeneration and death of DA neurons. The importance of *Debcl*-induced apoptosis is exhibited by the strict control in its gene product by the tumour suppressors *Rbf1* (*Clavier et al., 2015*), *OSCP1/NOR1* (*Huu, Yoshida & Yamaguchi, 2015*), and *NF-Y* (*Ly et al., 2013*). Furthermore, it has a motif for ubiquitination, probably by the *TrCP* homologue *slimb* that targets it for destruction by the proteasome (*Colin et al., 2014*). The inhibition of *Debcl* had a converse result, with flies that had a longer lifespan and healthy climbing ability. It is possible that the suppression of *Debcl* tips the balance towards the survival pathways controlled by the antiapoptotic *Buffy*. Our results indicate that overexpression of *Debcl* appears to be a novel model of PD as a result of neuronal apoptosis.

The α-synuclein-induced model of PD in Drosophila shows little difference in lifespan between the control and wild type, A53T and A30P α-synuclein flies (*Feany & Bender, 2000*). In our study, the overexpression of *Debcl* in the DA neurons resulted in a marked decrease in lifespan. This is in part due to toxic effects as a result of the expression of α-synuclein, and additionally, due to *Debcl*-induced apoptosis. The *Debcl*-induced apoptosis is mediated by other factors including; the mitochondrial fission protein *Drp1* (*Clavier et al., 2015*) that interacts with *Debcl* to induce mitochondrial fragmentation; *Glycerophosphate oxidase-1* (*Colin et al., 2015*) that increases mitochondrial ROS accumulation; and possibly through the initiation of autophagy, since both α-synuclein expression (*Xilouri & Stefanis, 2015*) and *Debcl* (*Hou et al., 2008*) overexpression are implicated in this process. This worsening of phenotypes was also observed when *Debcl* was overexpressed with α-synuclein in the eye. The inhibition of *Debcl* in the DA neurons resulted in a marked increase in survival and improved locomotor ability. This inhibition of *Debcl* is sufficient to negate its apoptotic role and thus promote cell survival through the opposing antiapoptotic *Buffy*.

Locomotor dysfunction is one of the major symptoms of PD. The demonstration of an age-dependent loss of climbing ability is pivotal to highlighting the effects of degeneration and death of DA neurons, ultimately as a consequence of altered gene expression as opposed to cellular senescence (*Rodriguez et al., 2015*). The overexpression of *Debcl* in the DA neurons produced a climbing index significantly different from that of control flies with the loss of climbing ability in an age-dependent manner and likely due to *Debcl*-induced neuronal degeneration. The degree of locomotor dysfunction seemed to be similar to that observed when α-synuclein is overexpressed in DA neurons. Taken together, these results would indicate a detrimental effect in overexpression of *Debcl* in DA neurons that result in a novel model of PD in flies.

In contrast, the inhibition of *Debcl* in the same neurons results in a remarkable improvement in climbing ability when compared to the controls. The inhibition of *Debcl* in the DA neurons of the α-*synuclein*-induced PD model significantly increased lifespan and climbing ability, indicating that reduced levels of *Debcl* are sufficient to alter the healthspan of DA neurons. The *Debcl*-induced apoptosis relies on downstream effectors that either induces ROS accumulation (*Colin et al., 2015*) or the fragmentation of the mitochondria (*Clavier et al., 2015*). As the down-regulation of *Buffy* or up-regulation of *Debcl* results in apoptosis (*Huu, Yoshida & Yamaguchi, 2015*), the cellular advantage of *Debcl* inhibition may be indirect through the de-repression of the *Buffy* gene product that confers survival advantages. The directed expression of *Buffy* along with *Debcl* results in an improved "healthspan" compared to the *Debcl*-induced phenotypes and corroborate other studies that show the overexpression of the pro-survival *Buffy* confers survival advantages through increased survival and improved climbing ability under conditions of stress (*M'Angale & Staveley, 2016*). Our study suggests that the overexpression of *Buffy* is similar to an up-regulation that ultimately blocks *Debcl*-induced apoptosis, similar to results obtained when its regulation by Rbf1 or dE2F2 is altered to repress it transcriptionally (*Clavier et al., 2014*; *Clavier et al., 2015*). This suppression of *Buffy* is sufficient to induce *Debcl*-dependent apoptosis, in addition to the promotion of *Debcl* activity by dNF-Y (*Ly et al., 2013*). The co-overexpression of *Debcl* and *Buffy* in the eye resulted in a partial rescue of the *Debcl*-induced phenotypes. Therefore, overexpression of the pro-survival *Buffy* suppresses the *Debcl*-dependent phenotypes.

## CONCLUSIONS

Directed inhibition of *Debcl* results in improved survivorship and extended climbing ability whereas the directed expression of *Debcl* results in reduced lifespan and impaired locomotor function. These phenotypes are rescued upon co-expression with the pro-survival *Buffy*. The overexpression of *Debcl* enhances the effects of α-*synuclein* expression. *Buffy* counteracts *Debcl*-induced phenotypes, and represents a potential target to enhance neuronal survival in response to the detrimental effects of *Debcl*-induced apoptosis.

### Funding

PGM was partially funded by the Department of Biology Teaching Assistantships and a School of Graduate Studies Fellowship from Memorial University of Newfoundland. BES was funded by the Natural Sciences and Engineering Research Council of Canada (NSERC) Discovery Grant. The funders had no role in study design, data collection and analysis, decision to publish, or preparation of the manuscript.

### Grant Disclosures

The following grant information was disclosed by the authors:
Department of Biology Teaching Assistantships and a School of Graduate Studies

Fellowship from Memorial University of Newfoundland.
Natural Sciences and Engineering Research Council of Canada (NSERC)
Discovery Grant.

## Competing Interests
The authors declare that they have no competing interests.

## Author Contributions
- P. Githure M'Angale conceived and designed the experiments, performed the experiments, analyzed the data, contributed reagents/materials/analysis tools, wrote the paper, prepared figures and/or tables, reviewed drafts of the paper.
- Brian E. Staveley conceived and designed the experiments, contributed reagents/materials/analysis tools, wrote the paper, reviewed drafts of the paper.

## Data Deposition
    The raw data has been supplied as Supplemental Dataset Files.

## Supplemental Information
Supplemental information for this article can be found online at http://dx.doi.org/10.7717/peerj.2461#supplemental-information.

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
