# Peer review of "Bcl-2 homologue Debcl enhances α-synuclein-induced phenotypes in Drosophila"

_PeerJ, doi:10.7717/peerj.2461_

## Round 0.1 · original submission · Major Revisions

· Academic Editor

Major Revisions

The paper needs a major revision.

In addition to the comments from the reviewers, I would like to especially point out the quality of the eye pictures in figure 4. Also, there is no quantification of the data, and no specification of the actual numbers, sex, etc. of the flies used for the eye experiments. Please include these data in your revised version, if you choose to do so, and a quantification of the data and better quality pictures of the different eye genotypes, together with a more detailed explanation of the methods used.

Please also refer to your recent paper published this last month in BMC Neuroscience, and since the data is germane to the matter of this paper, please discuss the results in that paper together with the data presented in this manuscript.

Reviewer 1 ·

Basic reporting

No comments

Experimental design

The manuscript “Bcl-2 homologue debcl enhances α-synuclein-induced phenotypes in Drosophila” intends to demonstrate that an induction of the apoptotic phenotype induced by a proapototic mitochondrial homologue of Bcl2 enhances α-synuclein phenotypes. The problem with this approach is that the sole expression of debcl in the eye has such an extreme phenotype that it suggests that the cells that overexpress debcl are already very close to become apoptotical and therefore any further stressor could be sufficient to enhance this phenotype in a non-specific manner. In other words α-synuclein would be a non-specific enhancer of debcl instead of debcl being an specific enhancer of α-synuclein. Although this conclusion superficially seems to be semantic, it is not so, since it has been proven that mutations in α-synuclein do cause familial Parkinsonism while debcl modulates apoptosis, a “normal” cellular phenomenon, therefore the experiments presented are more about the effects of the debcl over expression than about the α-synuclein-induced phenotypes. I think the manuscript should be modified in order to reflect this change in the logical interpretation of the results.
The other mayor problem that I see with this work is that the “rough eye” phenotype induced by the expression of α-synuclein is a lot more penetrant than the observed in other reports where α-synuclein is expressed in the eye, this stronger than normal α-synuclein phenotype should be, if possible, explained or some other example in previous literature should be noted. The enhanced “rough eye” induced by α-synuclein could be caused by the genetic background where the experiments are done or by some other uncontrolled circumstance, therefore “rough eye “ phenotypes of all the genetic combinations used in this work should be quantified in the population in order to be able to really compare genetic enhancement or suppression. Cuantitation should be done using an objective method such as: Diez-Hermano, S., Valero, J., Rueda, C., Ganfornina, M. D. & Sanchez, D. An automated image analysis method to measure regularity in biological patterns: a case study in a Drosophila neurodegenerative model. Mol. Neurodegener. 10, 1–10 (2015).

The behavioral and survival experimental curves are interesting and informative but the differences between the different phenotypes are hard to compare as they are presented. The figures should be modified in order to make them easier to compare. Maybe, the addition of another panel with histograms showing the moment at where half of the population is dead could be added in order to compare differences between populations. Similarly the climbing index of all genetic combinations used or quantified should be compared with the control´s at the time half of the control population is still alive.

Validity of the findings

The stronger than normal α-synuclein phenotype should be, if possible, explained or some other example in previous literature should be noted. The enhanced “rough eye” induced by α-synuclein could be caused by the genetic background where the experiments are done or by some other uncontrolled circumstance, therefore “rough eye “ phenotypes of all the genetic combinations used in this work should be quantified in the population in order to be able to really compare genetic enhancement or suppression. Cuantitation should be done using an objective method such as: Diez-Hermano, S., Valero, J., Rueda, C., Ganfornina, M. D. & Sanchez, D. An automated image analysis method to measure regularity in biological patterns: a case study in a Drosophila neurodegenerative model. Mol. Neurodegener. 10, 1–10 (2015).

Reviewer 2 ·

Basic reporting

This work fulfills most of the criteria required for publishing in PeerJ.
It is written in clear, unambiguous, professional English language used throughout.
The intro and background shows context appropriately. The literature is relevant and it is well referenced but the reference section must be reviewed for formatting and complete the information in many of the works cited.
The structure of the paper is well designed and clear.
The figures are relevant and of high quality, well labeled and described.

Experimental design

The research in this work is within the scope of the journal.
The question is well defined, relevant and meaningful and it clearly points out the goal of the investigation an identified the knowledge gap.
Rigorous investigation is performed.
Some methods should be described with more detail. The major experimental design they are using to approach its question, is the overexpression of UAS-transgenes or knocking down of the proteins of interest with UAS-RNAis with the bipartite system GAL4-UAS. They use drivers to direct the expression of these constructs to dopaminergic neurons and they evaluate mostly three phenotypes: survival, climbing activity and structure of the adult eye.
I think they should describe or reference with more clarity the structure of the transgenes and RNAis used in this work and the strategies they used to keep track of the recombination events they did to get the genotypes used.

Validity of the findings

The authors get clear results. Data are robust, statistically sound and controlled.
The conclusion is well stated, linked to original research question and limited to supporting results.

Additional comments

This paper is well presented and written. The reasoning is correct and well conceived.
It is a continuation of the findings published in M´Angale and Staveley, 2016 (BMC Neurosci. 2016 May 18;17(1):24. doi: 10.1186/s12868-016-0261-z). They performed a quite similar study with Buffy and found complementary conclusions. I wonder why they publish these results in different papers with such a short difference in time.

It will be interesting to study in debcl-knockdown flies whether the rescue of all the phenotypes evaluated are due to the reasons the authors propose such as the derepression of Buffy. I think they have a nice model to perform more experiments to understand some aspect of neurodegeneration.
I think this paper could be accepted once they revise the following points:

MAJOR POINTS

1. Proof that the debcl and Buffy RNAis are knocking down the expression of these proteins either by checking the correspondent mRNAs by, for example RT-PCR, or with Western analyses of each protein. This will give an indication to whether the antagonism they are observing between the activities of these proteins, is due to a similar knocking down effect of each protein.

2. The statement “The importance of debcl is demonstrated by the different promoters contained in its genomic regions including the 5´ nuclear transcription factor Y (NF-Y) which has been shown to be important for gene promoter activity (Ly et al. 2013)” (lanes 108-110) is completely incorrect. I think it mixes the concepts of promoters, regulatory elements and transcription factors. This should be corrected and clarified.

3. In their analyses of functional sites of Debcl, they mentioned that they found a transmembrane domain, an inhibitor of apoptosis binding motif (IBM), a PDZ domain, an ER retention motif, an Atg8 binding motif, a nuclear receptor box motif, and a ubiquitination motif. All these motifs should be highlighted in Figure 1 (they are not).


MINOR POINTS

1. Many references are incomplete or have formatting mistakes or missing information. Thorough review of this section most be made before considering for publishing. Only some examples are:

Ambegaokar SS, Roy B, and Jackson GR. 2010. Neurodegenerative models in Drosophila: polyglutamine disorders, Parkinson disease, and amyotrophic lateral sclerosis. Neurobiology of disease 40:29-39.
Gasser T. 2009. Molecular pathogenesis of Parkinson disease: insights from genetic studies. Expert Reviews in Molecular Medicine 11:null-null.
Guo M. 2012. Drosophila as a model to study mitochondrial dysfunction in Parkinson's disease. Cold Spring Harbor perspectives in medicine 2.
Gupta A, Dawson VL, and Dawson TM. 2008. What causes cell death in Parkinson's disease? Annals of Neurology.
Kong Y, Liang X, Liu L, Zhang D, Wan C, Gan Z, and Yuan L. 2015. High Throughput Sequencing Identifies MicroRNAs Mediating alpha-Synuclein Toxicity by Targeting Neuroactive-Ligand Receptor Interaction Pathway in Early Stage of Drosophila Parkinson's Disease Model. PLoS One 10:e0137432.
Jörg BS. 2007. Mechanisms of neurodegeneration in idiopathic Parkinson's disease. Parkinsonism & Related Disorders 13.
Staveley BE. 2014. Drosophila Models of Parkinson Disease. In: LeDoux MS, ed. Movement Disorders: Genetics and Models. Second ed: Elsevier Science, 345-354.
Webb JL, Ravikumar B, Atkins J, Skepper JN, and Rubinsztein DC. 2003. Alpha-Synuclein is degraded by both autophagy and the proteasome. The Journal of biological chemistry 278:25009-25013.
Zhu ZJ, Wu KC, Yung WH, Qian ZM, and Ke Y. 2016. Differential interaction between iron and mutant alpha-synuclein causes distinctive Parkinsonian phenotypes in Drosophila. Biochimica et Biophysica Acta (BBA) - Bioenergetics.

2. I think the correct form for Debcl protein in Drosophila is with a capital letter at the beginning. I would suggest correcting it through all the manuscript.

3. In lane 123, they talk about a MOM-targeting motif. What MOM is should be stated.

4. In the Material and Methods section, the authors stated that to generate the lines for overexpression of either alpha-synuclein or Buffy in DA neurons used in this work, they used standard homologous recombination methods and that these events were corroborated by PCR reactions. They should detail why and what sequences they recombined and with what primers they analyzed the sequences to corroborate the recombination events.

5. Lane 162. Drosophila should start with a capital letter.

6. Figure 2B legend: “the overexpression of debcl…” debcl should be in italics.

7. Lane 227: “overexpression of debcl…” debcl should be in italics.
8. Lane 282: “overexpression of debcl…” debcl should be in italics.

9. Lane 294: M’Angale & Staveley, in press. Should be added to references. BMC Neurosci. 2016 May 18;17(1):24. doi: 10.1186/s12868-016-0261-z.

---

## Round 0.2 · Major Revisions

· Academic Editor

Major Revisions

Please consider answering the comments expressed by reviewers, especially reviewer #2, in a newly revised version of the manuscript.

Reviewer 1 ·

Basic reporting

No comments

Experimental design

No comments

Validity of the findings

The penetrance of the rough eye phenotype caused by the expression of alfa-synuclein is really astonishing. Even if the flies were grown at 29 degrees centigrade the authors should support this result with an appropriate reference where similar phenotypes are caused by the sole over expression of alfa-sinuclein regardless of the temperature that is attained.

Reviewer 2 ·

Basic reporting

No Comments

Experimental design

No comments

Validity of the findings

No comments

Additional comments

I still think this manuscript needs a few major revisions to be accepted.
Reviewer 2 Comment 2
Experimental design
The research in this work is within the scope of the journal.
The question is well defined, relevant and meaningful and it clearly points out the goal of the investigation an identified the knowledge gap.
Rigorous investigation is performed.
Some methods should be described with more detail. The major experimental design they are using to approach its question, is the overexpression of UAS-transgenes or knocking down of the proteins of interest with UAS-RNAis with the bipartite system GAL4-UAS. They use drivers to direct the expression of these constructs to dopaminergic neurons and they evaluate mostly three phenotypes: survival, climbing activity and structure of the adult eye.
I think they should describe or reference with more clarity the structure of the transgenes and RNAis used in this work and the strategies they used to keep track of the recombination events they did to get the genotypes used.

Reviewer 2 Response 2
The RNAi lines/stocks used in this experiments were sourced from Bloomington Drosophila Stock Center. As to the derivative lines that we developed using standard recombination techniques, we tested with PCR. We will include the primers used for the detection of α-synuclein.

ANSWER TO REBUTTAL 2

The primers description for the PCR experiments is good but to finish clarifying this section, they should explain what they recombined and why, to obtain what particular genotype(s). What they did select?, etc.

Reviewer 2 Comment 6
I think this paper could be accepted once they revise the following points:
MAJOR POINTS
1. Proof that the debcl and Buffy RNAis are knocking down the expression of these proteins either by checking the correspondent mRNAs by, for example RT-PCR, or with Western analyses of each protein. This will give an indication to whether the antagonism they are observing between the activities of these proteins, is due to a similar knocking down effect of each protein.

Reviewer 2 Response 6
We have discussed this in our lab, for the current paper, this was not within the scope of our goals of the study, in future though, this will be an important supporting study. Importantly to note though is that the RNAi lines are designed by other researchers whose findings on expression patterns is available on FlyBase and other sources, and we only apply the lines to extend previous studies.

ANSWER TO REBUTTAL 6
If so, at least authors should recopilate that information and cite the sources. I still think that RT-PCRs are not complicated or expensive and that would improve the importance of their conclusions.

Reviewer 2 Comment 7
2. The statement “The importance of debcl is demonstrated by the different promoters contained in its genomic regions including the 5´ nuclear transcri9ption factor Y (NF-Y) which has been shown to be important for gene promoter activity (Ly et al. 2013)” (lanes 108-110) is completely incorrect. I think it mixes the concepts of promoters, regulatory elements and transcription factors. This should be corrected and clarified.

Reviewer 2 Response 7
Thank you, we have made the necessary corrections in the text to clarify this, deleted “different promoters contained in its genomic regions including”.

ANSWER TO REBUTTAL 7

The new phrase:
"The importance of Debcl is perhaps demonstrated by the presence of 5’ nuclear transcription factor Y (NF-Y) promoter region which has been shown to be important for gene promoter activity (Ly et al. 2013)" is still incorrect. It should be revised again.


MINOR POINTS
Reviewer 2 Comment 9
1. Many references are incomplete or have formatting mistakes or missing information. Thorough review of this section most be made before considering for publishing.

Reviewer 2 Response 9
Thank you for your keen observation, we use Endnote for citations and at times it returns errors, we will now and in future be more diligent. We have made the necessary changes and corrections.

ANSWER TO REBUTTAL 9
There are still mistakes in refs, e . g.

Colin J, Gaumer S, Guenal I, Mignotte B. 2009. Mitochondria, Bcl-2 family proteins and apoptosomes: of worms, flies and men. Frontiers in bioscience 14:4127-4137

Dinkel H, Van Roey K, Michael S, Kumar M, Uyar B, Altenberg B, Milchevskaya V, Schneider M, Kuhn H, Behrendt A, Dahl SL, Damerell V, Diebel S, Kalman S, Klein S, Knudsen AC, Mader C, Merrill S, Staudt A, Thiel V, Welti L, Davey NE, Diella F, Gibson TJ. 2016. ELM 2016-data update and new functionality of the eukaryotic linear motif resource. Nucleic acids research 44:D294-300

etc.

---

## Round 0.3 · Major Revisions

· Academic Editor

Major Revisions

Please heed the comments from reviewer#2; there are still extant issues that have not been answered in the previous two versions, so in order to move forward with this submission those concerns should be addressed.

Reviewer 2 ·

Basic reporting

No comments

Experimental design

No comments

Validity of the findings

No comments

Additional comments

My previous review had three major points that I consider are not clarified in this new version.

1. The sentence in this version "The importance of Debcl is perhaps demonstrated by the presence of 5’ nuclear transcription factor Y (NF-Y) promoter region which has been shown to be important for gene promoter activity (Ly et al. 2013). " is still incorrect!!

Please do check and understand the reference cited.

2. In Material and Methods they describe in this new version:

"The UAS-α-synuclein/CyO; Ddc-Gal4/TM3; UAS-α-synuclein/CyO; GMR-Gal4; UAS-Buffy/CyO; Ddc-Gal4 and UAS-Buffy/CyO; GMR-Gal4 derivative lines were generated using standard homologous recombination methods that have been previously described (Rong & Golic 2000). But briefly, the UAS construct and the driver are crossed and the progeny is crossed to a balancer in our case lobbed eyes and CyO, the progeny is then selected and tested by PCR.

I still do not see what they exactly recombined by homologous recombination. Why they have to do this?? Please describe carefully what has been done.

3. The references section still have many mistakes.

---

## Round 0.4 · accepted · Accept

· Academic Editor

Accept

As you can see, Reviewer 2 still has comments about the references. However, the Editorial Office has assured me that these issues can be resolved while in Production, so I am happy to Accept the paper at this point.

Reviewer 2 ·

Basic reporting

No comments

Experimental design

No comments

Validity of the findings

No comments

Additional comments

Finally looks that it is getting better. There are still mistakes in the references.